# Multifaceted Roles of Aquaporins in the Pathogenesis of Alzheimer’s Disease

**DOI:** 10.3390/ijms24076528

**Published:** 2023-03-31

**Authors:** Kaoru Yamada

**Affiliations:** Department of Neuropathology, Graduate School of Medicine, The University of Tokyo, Tokyo 113-0033, Japan; yamadaka@m.u-tokyo.ac.jp

**Keywords:** aquaporins, ISF, CSF, Alzheimer’s disease, glymphatic system, Aβ, tau

## Abstract

The central nervous system is highly dependent on water, and disturbances in water homeostasis can have a significant impact on its normal functions. The regulation of water balance is, at least in part, carried out via specialized water channels called aquaporins. In the central nervous system, two major aquaporins (AQPs), AQP1 and AQP4, and their potential involvements have been long implicated in the pathophysiology of many brain disorders such as brain edema and Neuromyelitis optica. In addition to these diseases, there is growing attention to the involvement of AQPs in the removal of waste products in Alzheimer’s disease (AD). This indicates that targeting fluid homeostasis is a novel and attractive approach for AD. This review article aims to summarize recent knowledge on the pathological implications of AQPs in AD, discussing unsolved questions and future prospects.

## 1. Introduction

Brain interstitial fluid (ISF) and cerebrospinal fluid (CSF) are two types of extracellular fluid that support the central nervous system (CNS). ISF surrounds and fills the spaces between brain parenchyma cells and constitutes 12–20% of the brain fluid [1]. In contrast, CSF is an extracellular fluid distinct from ISF that surrounds the brain and spinal cord, filling the ventricles of the brain and the subarachnoid space, accounting for 10% of the intracranial water volume [1]. These extracellular fluids not only provide a protective cushioning effect for the brain but also play crucial roles in the transport of nutrients and waste products, maintenance of electrolyte balance, and signal transduction. Water homeostasis in these compartments is regulated by specialized water channels called aquaporins (AQPs).

Dysregulation in AQPs has been linked to a range of neurological conditions and is a target for the treatment of brain diseases, such as brain edema, ischemia and Neuromyelitis optica [2].

In addition to these diseases, the roles of AQPs in the removal of waste products in Alzheimer’s disease (AD) in particular has received significant scientific attention in recent years. This article not only gives a general introduction to AD (Section 2 and Section 3) but also aims to provide the readers with knowledge of the molecular structure and localization of AQPs (Section 4), their physiological functions and pathological implications (Section 5, Section 6, Section 7, Section 8 and Section 9), as well as discussing the limitations of current knowledge (Section 10).

## 2. AD

AD is the most common cause of dementia, with an estimated prevalence of 50 million cases worldwide. AD is pathologically characterized by the extracellular accumulation of beta-amyloid (Aβ) plaques and intracellular neurofibrillary tangles in the brain, as well as by the loss of neurons and synapses. Aβ is a peptide derived from an amyloid precursor protein (APP) expressed in neurons through a series of enzymatic cleavage by β-secretase and γ-secretase, and it form aggregates known as amyloid plaques (Figure 1). The length of the Aβ peptide varies, with Aβ40 and Aβ42 being the most common forms. Aβ40 is more abundant, but Aβ42 is more prone to aggregate and is believed to be more toxic. Neurofibrillary tangles are intracellular aggregates composed of the abnormally phosphorylated microtubule-binding protein tau (hereinafter referred to as p-tau), which is also predominantly expressed in neurons (Figure 1). Unlike sporadic AD, which accounts for more than 95% of all cases, familial Alzheimer’s disease (FAD) is a rare genetic form of AD that is inherited in an autosomal dominant pattern. There are three known genes that are associated with FAD: amyloid precursor protein (APP), presenilin 1 (PSEN1), and presenilin 2 (PSEN2). Mutations associated with FAD affect the cleavage of APP by β-secretase and/or γ-secretase, leading to an increase in the total production of Aβ or Aβ42 relative to Aβ40, suggesting its key role in the development of AD. The clinical trials of aducanumab, a drug recently approved by the FDA, suggests that targeting Aβ may be a viable therapeutic approach for AD.

The accumulation of Aβ is among the earliest pathological changes in the brain of individuals with AD. Longitudinal data obtained from familial AD cases reveal that AD starts approximately 20 years before the onset of clinical symptoms, followed by an increase in neurofibrillary tangle deposition and neurodegeneration [3]. 

In contrast to Aβ, genetic mutations in the tau gene (*MAPT*) have not been identified in AD, yet its association with several rare inherited diseases accompanied with tau aggregation in neurons and glial cells (referred to as tauopathies) strongly indicates that tau abnormality contributes to the degeneration of neurons. Consistent with this, the distribution of neurofibrillary tangles is closely associated with cognitive impairment in AD.

Genetic analysis of FAD cases, as well as animal studies, revealed that the extracellular deposition of Aβ triggers a series of downstream events, including tau deposition, neuroinflammation, and neurodegeneration. Transgenic mouse models that recapitulate the two pathological features of AD have been used to study its underlying mechanisms and test for potential treatments. These include several lines of APP-transgenic/knock-in mice harboring FAD-linked mutations that develop age-dependent Aβ plaques in the brain [4,5], as well as mutant tau transgenic mice that deposit neurofibrillary tangles and exhibit neurodegeneration [6,7].

In the process of aggregation, a “seed”—a small, oligomeric form or misfolded protein—is generated and acts as a nucleation site for further accumulation and aggregation. The aggregation of these aberrant proteins is believed to occur in a concentration-dependent manner. Therefore, as the concentration of these proteins in the brain increases, the likelihood of their aggregation also increases. This means that the reduced efficiency of the clearance mechanisms could be one cause of the buildup of these pathogenic proteins in the brain. 

Growing evidence suggests that impaired clearance mechanisms play a specific role in the development and progression of Aβ pathology in AD. One line of evidence supporting this idea comes from a study that used stable isotope labeling and mass spectrometry. In this study, the authors demonstrated that sporadic late-onset forms of AD cases have a decreased rate of Aβ clearance in CSF [8], while FAD is linked to an increased rate of Aβ production. The clearance of Aβ or tau can be mediated through a combination of multiple intracellular or extracellular mechanisms, including enzymatic degradation and transport across the blood–brain barrier, autophagy–lysosomal pathway, ubiquitin–proteasome system and glymphatic system. The deficits in these pathways could contribute to the increase in AD pathology.

## 3. Extracellular Aβ and Tau

As CSF can be exchanged with ISF, which is in closer proximity to neurons, it reflects the changes occurring in the brain, at least to some degree. CSF biomarkers can provide valuable information regarding the underlying pathological changes that occur in the brains of patients with AD. The most commonly used biomarkers in AD are CSF Aβ42 and total tau or p-tau. In people with AD, CSF levels of Aβ42 are lower and inversely correlated with Aβ plaque load, likely due to the sequestration of Aβ42 to the plaques. In contrast, the levels and phosphorylated forms of tau are higher than those in healthy individuals. It has been considered that this elevation of tau in the CSF of patients with AD reflects the degree of neuronal loss, meaning that more tau is passively leaked from damaged or degenerating neurons in AD. However, accumulating data now challenge this assumption. First, the rate of increase in CSF tau and p-tau may actually become slower when neurodegeneration increases [9], and the production rate of tau is instead increased in the presence of Aβ plaques [10]. Second, cellular and animal studies have demonstrated that tau is normally released into the extracellular space, for example, ISF under physiological conditions [11,12,13], and several mechanisms responsible for its release have been identified (reviewed in [14]). Consistent with this, it has been shown that the presence of the Aβ plaque is sufficient for inducing an increase in CSF total tau and p-tau levels without neurodegeneration [15,16].

The pathological significance of extracellular tau is also supported by transcellular propagation of tau aggregates. In addition to soluble, monomeric forms of tau, aggregation seeds can also exit and transmit neurons via the extracellular space [17,18]. The detection of tau seeding activity in human CSF also supports the propagation hypothesis [19].

These observations collectively indicate that not only Aβ but also tau can have connections with extracellular fluid homeostasis pathways.

## 4. Molecular Structure and Localization of AQP4

Aquaporins (AQPs) are a family of trans-membrane proteins that function as channels selectively permeable to water across cell membranes. The structure of AQPs is highly conserved across different types and species, reflecting the critical role of these proteins in maintaining water balance in cells and tissues.

AQP4 is the most abundant water channel in the brain followed by AQP1 and AQP9. There are two major isoforms of AQP4: AQP4-M1 and AQP4-M23. AQP4-M1 and AQP4-M23 are two different splice variants of the AQP4 gene that result from alternative splicing of the mRNA transcript. AQP4-M1 is a longer isoform that is generated by translation initiation at Met1. AQP4-M23, on the other hand, is a shorter isoform that is generated by translation initiation at the Met23 position. Both AQP4-M1 and AQP4-M23 isoforms connect in membranes as tetramers and form supramolecular assemblies called orthogonal arrays of particles (OAPs) [20]. OAP size changes depending on the ratio of AQP4-M1 to AQP4-M23, although the functional consequences of this change are uncertain.

AQP4 is highly enriched in perivascular astrocytic endfoot processes that ensheath cerebral blood vessels. This polarized expression of AQP4 at perivascular astrocytes is caused by interactions with a dystrophin-associated complex, including α-syntrophin, dystrobrevin, dystophin, and dystroglycan, which anchor AQP4 to the extracellular matrix. It has been shown that these components impact the cellular localization of AQP4. A study has demonstrated that α-syntrophin knockout mice showed a marked reduction in AQP4 from the perivascular regions without altering their expression in other membrane domains [21]. The other component that contributes to the localization of AQP4 is agrin, a heparan sulfate proteoglycan found in the extracellular matrix [22].

AQP4 localization is also controlled by various intracellular mechanisms, including phosphorylation by protein kinases and actin reorganization [23]. For example, AQP4 relocalizes to the plasma membrane from intracellular vesicular pools in response to hypotonic stimuli, which is regulated by the activation of PKA and calmodulin [24,25]. In addition to these post-translational modifications, AQP4 localization is also regulated at a translational level. An AQP4 splice variant lacking exon 4 has been identified in humans; it exerts a dominant-negative effect on plasma membrane expression and water permeability [26]. Programmed translational read-through is another regulatory mechanism. Translation usually terminates at an in-frame stop codon; however, in certain cases, translation continues beyond the canonical stop codon. This generates polypeptides with C-terminal extensions that often acquire new properties or subcellular localizations. In the case of AQP4, the 29-amino-acid C-terminally elongated variant arising from the translational readthrough is called AQP4-ex or AQP4-x. Although water transport properties are not substantially altered by the C-terminal extension, the localization of AQP4-x is exclusively perivascular within the astrocytic endfoot, likely due to its stronger C-terminal interaction with its binding partner, α-syntrophin [27]. A recent study also demonstrates that AQP4-x is primarily responsible for glymphatic clearance, as described in the following section.

## 5. Glymphatic Clearance and Its Implications in AD

Knowledge on the physiological functions of AQP4 in the CNS is largely based on the phenotypes of AQP4 knockout mice [28,29,30]. AQP4 deficiency in mice does not produce major structural abnormalities or behavioral phenotypes, except for a moderate increase in water content. Thus, its functions under physiological conditions remained uncertain until its role in extracellular fluid flow was discovered.

The physiological functions of AQP4 in the CNS have been extensively studied, mostly in the context of the glymphatic system. It was discovered because of its dependence on astroglial cells and its functional similarity to peripheral lymphatic system [31]. The glymphatic system is a recently discovered waste clearance system that is responsible for removing metabolites and waste products produced in the brain. In this model, CSF from the subarachnoid spaces flows through the perivascular spaces of the penetrating arteries and mixes with ISF containing waste and metabolites, and then drains out through the perivenous spaces [31] (Figure 2). During this circulation, CSF is absorbed by the dural lymphatic vessels and transported to the cervical lymph nodes [32,33]. Although there remains some controversy as to whether it is actually mediated by convective flows rather than diffusion, the discovery of glymphatic clearance has significantly increased the interest in brain fluid transport.

Although glymphatic clearance was first discovered and characterized in the rodent brain, subsequent MRI studies have shown that it is also present in humans [34,35]. An arterial pulsation is one of the driving forces of the glymphatic convective flow [36,37]. In addition, AQP4 is also critical for the proper functioning of the glymphatic system. The requirement of AQP4 in the glymphatic system has been confirmed in multiple lines of AQP4 KO mice, as well as pharmacological blockade by the AQP4 inhibitor, TGN-020 [38,39]. It has recently been shown that AQP4 KO mice display an enlarged interstitial volume and decreased CSF volume without an altered CSF production rate [40]. The stagnation of ISF and enlargement of the ISF space might be a consequence of an overall reduction in glymphatic transport.

The loss of the perivascular AQP4 expression correlates with decreased glymphatic clearance. Studies have shown that glymphatic clearance is suppressed in α-syntrophin KO mice, which reduces perivascular AQP4 levels [21]. Another piece of evidence suggesting the pivotal role of perivascular AQP4 in the glymphatic system comes from a study of the translational readthrough variant, AQP-x. It has been shown that AQP4-x-specific knockout mouse lines generated by CRISPR-Cas9 elevate ISF Aβ levels with a significant increase in the half-life [41]. Collectively, these lines of evidence suggest a crucial role of the specific localization of AQP4 in perivascular astrocytes in glymphatic clearance.

The spatial and temporal patterns of AQP are also regulated by the circadian rhythm, which generates a dynamic day/night difference in glymphatic flow [42].

Although many experiments have relied on AQP4 KO mice to study glymphatic function, the molecular mechanisms underlying how water transport through AQP4 alters extracellular fluid flow remain unclear.

## 6. The Impact of Glymphatic Clearance on AD Pathogenesis

The impairment of glymphatic clearance has been implicated in aged brains and many neurodegenerative diseases [43,44]. Reduced glymphatic clearance was observed in aged mice with a concomitant reduction in perivascular AQP4 expression. Consistent with these findings, altered fluid dynamics have been reported in AD patients. A study using positron emission tomography (PET) following an intravenous injection of [^15^O]H_2_O revealed that water influx into CSF was significantly reduced in AD patients compared to controls [45]. An imaging study using MRI also revealed a reduction in water influx into CSF in APP-transgenic mice [46]. Genetic analysis has also demonstrated that single-nucleotide polymorphisms in the non-coding region of AQP4 are associated with an altered rate of cognitive decline after the diagnosis of AD [47], suggesting that alteration of the glymphatic system might be a causative factor for development of AD.

To address the potential causal link between glymphatic impairment and AD pathogenesis, the effects of AQP4 deletion on AD pathology were investigated using several lines of AD model mice. Studies have found that mice lacking AQP4 show aggravated Aβ deposition [48,49], cognitive function, and epileptiform neuronal activity in a mouse model of Aβ pathology [50].

Tau is also released into ISF and is thus a potential substrate for extracellular clearance pathways. Reducing the extracellular levels of pathogenic tau might be beneficial, as anti-tau antibodies could inhibit the transcellular propagation of tau by binding to extracellular tau aggregates, preventing their uptake by neighboring cells and reducing the spread of tau pathology [51]. However, even in the absence of anti-tau antibodies, there should be intrinsic mechanisms in CNS that clear extracellular tau from ISF [52], which were not described until recently. Based on the hypothesis that the glymphatic system may be involved in extracellular tau clearance, we and other groups have recently demonstrated that tau clearance into CSF is significantly suppressed by the genetic deletion of AQP4 or AQP4 inhibitors [39,53]. More importantly, the deletion of AQP4 in tau transgenic mice not only elevated CSF tau levels but also strongly exacerbated tau pathology and neuronal loss, suggesting that impaired glymphatic clearance might be one cause of tau-associated neurodegeneration [53]. This result is consistent with the previous observations that AQP4 deficiency accelerates the development of neurofibrillary tangle pathology and neurodegeneration in the post-traumatic brain [54]. The mechanisms by which the impairment of glymphatic tau clearance exacerbates tau pathology remain to be investigated. One possible explanation for this is that deficits in extracellular tau clearance facilitate the intercellular transmission of tau pathology. Another possibility is a potential equilibrium between the extracellular and intracellular compartments, where tau in both compartments could ultimately be modulated by glymphatic clearance [11].

Apolipoprotein E (apoE) is a component of lipoprotein particles and primarily involved in lipid metabolism. In CNS, apoE is primarily synthesized and secreted by astrocytes. There are three major apoE isoforms in humans called apoE2, apoE3, and apoE4, which differ from each other by single amino-acid changes at positions 112 and 158. While apoE3 isoform is the most common form of apoE, the apoE2 isoform is the rarest form of apoE, and it is associated with a lower risk of developing AD. Conversely, the apoE4 isoform is the strongest genetic risk factor identified to date for developing late-onset AD, and it is associated with an increased accumulation of Aβ in the brain as well as tau-associated neurodegeneration [55,56]. Interestingly, one study found that the glymphatic system also contributes to the delivery of CSF-derived apoE in an apoE-isoform-dependent manner [57].

While AQP4 deficiency elevated both Aβ and tau pathology, the accumulation of these aberrant proteins can in turn influence the expression or localization of AQP4. While AQP4 expression is highly polarized in wild-type mice, this polarized expression is disrupted in APP-transgenic mice with a significant increase in parenchymal AQP4 expression [49]. Abe et al. also reported that AQP4 expression was significantly elevated in reactive astrocytes surrounding Aβ plaques [50]. Although the functional consequences of this over-expression and its relationship with astrogliosis are uncertain, glymphatic flows are reduced in APP-transgenic mice [58]. A reduced polarized expression of AQP4 has also been reported in rTg4510 tau transgenic mice with a concomitant functional reduction in ISF and CSF exchange [39]. These observations indicate the presence of a vicious cycle between glymphatic clearance and AD pathology formation.

A study microinjecting synthetic Aβ has shown that Aβ directly interferes with the glymphatic inflow of CSF into the brain [58]. Glymphatic transport has also been associated with cerebral amyloid angiopathy (CAA). Cerebral amyloid angiopathy (CAA) is a condition characterized by the accumulation of Aβ in the walls of small- and medium-sized blood vessels in the brain and is commonly associated with AD. It has been shown that glymphatic transport is compromised in a mouse model of CAA [59]. Interestingly, one study found that evoked vascular reactivity is reduced in mice with CAA, which corresponds to a slower clearance rate from ISF [60].

Reduced perivascular AQP4 localization has been also demonstrated via histological analysis using human postmortem brain tissues, showing that this is associated with local Aβ pathology as well as phosphorylated tau pathology [61,62].

Another basis that links AD with glymphatic failure comes from the relationship with sleep. AD is often associated with sleep disturbances, and patients with AD often present with sleep disturbances and reduced sleep quality. The glymphatic system is shown to be more active in a state of sleep than in wakefulness, likely due to changes in extracellular space that modulate resistance to fluid flows [63]. This could explain why extracellular Aβ and tau levels in ISF are higher during wakefulness as a consequence of suppressed clearance [64,65].

Although these studies collectively suggest a strong association between AD and the glymphatic system, whether glymphatic impairment, including AQP4 mislocalization, is a cause or consequence of AD remains a matter of debate.

## 7. Impact on Lymphatic Drainage

The CNS has long been considered free from classical lymphatic vessels. However, the discovery of functional lymphatic vessels in dura mater by two independent laboratories significantly revised this classical view [32,33]. The glymphatic system is connected to vessels that drain CSF, immune cells, and other macromolecules and transport them to the cervical lymph nodes. Although the glymphatic and lymphatic drainage systems are distinct, evidence suggests that they may be interconnected. A previous study showed that the drainage of CSF to the lymph nodes is regulated by daily rhythms, which are abolished in AQP4 knockout mice [42]. Another study also demonstrated that tau drainage from CSF to the deep cervical lymph nodes is significantly suppressed in AQP4 KO mice [53]. It is unclear whether the alterations in AQP4 KO mice are dependent on previously unrecognized AQP4 functions or secondary changes due to the genetic deletion of AQP4.

## 8. Other Physiological Functions of AQP4

Recovery from neuronal excitation requires the rapid clearance of K+ ions from interstitial spaces and astrocytes are responsible for this process. AQP4 deficiency decreases the capacity to buffer K+ ions. The role of AQP4 in K+ homeostasis was also found in vivo. Prolonged K+ clearance was observed in α-syntrophin KO mice [66] as well as AQP4-deficient mice [67]. The functional interaction between AQP4 and the inwardly rectifying potassium channel Kir4.1 might explain this phenotype but the exact mechanistic link between K clearance and water transport via AQP4 is unclear (Figure 2).

Interestingly, studies have suggested that the release of Aβ and tau are regulated by neuronal excitation [12,68,69,70]. In addition, it has been demonstrated that chronic neuronal activation can aggravate Aβ and tau pathology [70,71]. Therefore, it is tempting to speculate that AQP4 deficiency can also impact the release of these pathogenic proteins (Figure 2).

Astrocyte migration is an important process during brain development, injury, and repair. It is known that the migration of astrocytes is regulated by various extracellular signals, such as growth factors and extracellular matrix proteins, as well as intracellular signaling pathways. It has been suggested that AQP4 may also play a role in regulating astrocyte migration [72,73]. One proposed mechanism is that AQP4 may regulate the formation and retraction of cell membrane protrusion at the leading edge of migrating astrocytes, although the mechanisms and their potential involvement in AD pathogenesis are not yet fully understood.

## 9. AQP1

AQP1 is the second most abundant water channel in the brain and is mainly found on the apical surface of the choroid plexus. AQP1 in the choroid plexus is involved in CSF secretion (Figure 2), which is important for maintaining proper volume and pressure of the CSF [74]. Although AQP1 has not been extensively studied in the context of AD, evidence suggests that it may play a role in the pathogenesis of the disease. There are studies showing that AQP1 expression is upregulated in reactive astrocytes surrounding Aβ plaques of AD patients [75,76]. Further research is needed to fully elucidate the role of AQP1 in AD and to determine its potential as a therapeutic target.

## 10. Conclusions

Studies have suggested that AQP4 plays a significant role in the pathogenesis of AD. In particular, its involvement in the glymphatic system is currently a very active and popular area of research in the field of neuroscience. However, there are still several unanswered questions in this field of research.

The crucial roles of AQP4 on AD pathogenesis are supported by the observation that a lack of AQP4 exacerbates AD pathology in animal models. However, as reviewed in this article, the roles of AQPs in the CNS are complex and multifaceted, and the exact mechanisms, as well as whether they influence AD pathogenesis, depend on the glymphatic system, which needs to be further explored.

Furthermore, most studies suggest that glymphatic clearance is suppressed in AD patients as well as in mouse models of AD. Yet, further studies are required to explore the mechanisms underlying this suppression. This includes the elucidation of the molecular and cellular basis of the reduced polarized expression of AQP4 in AD.

Although arterial pulsation and AQP4 are important factors in the regulation of glymphatic flow, many other factors can also influence this process as well. For example, extracellular tortuosity can affect the movement of substances through the extracellular space, which can be influenced by the extracellular matrix (ECM) structure, cellular morphology, and the interaction of solutes with cell membranes [77]. Of note, a recent study has demonstrated that parenchymal border macrophages regulate the CSF flow, possibly by altering ECMs [78]. Notably, CSF tracer influx was further decreased in AQP4-deficient APP/PS1 mice compared to that of AQP4 KO mice, suggesting that AQP4-independent failure may be involved [79]. Understanding the interplay between these various factors and their contributions to the regulation of the glymphatic system is also needed to better understand the precise mechanisms underlying the suppression of the glymphatic system in AD [78].

Another critical question from a therapeutic standpoint is whether the glymphatic system can be therapeutically targeted to slow or prevent the progression of AD. Although AQPs, especially AQP4 have emerged as a promising therapeutic target in AD, further research is required to determine the feasibility and effectiveness of this approach [80]. In addition to directly facilitating its channel function, regulating the trafficking of AQP4 to plasma membrane levels is another strategy. In fact, a recent study using high-throughput screening and counter-screening demonstrated that small molecule compounds that facilitate a translational readthrough of AQP4 can enhance the clearance of Aβ in vivo [41]. The development of suitable in vitro models to screen and validate the pharmacological regulation of AQP4 function will significantly facilitate translational research in this field. Further research is necessary to better understand the complex roles of AQPs in AD pathogenesis and to develop drugs that can effectively target AQP4 in the brain without disrupting other important functions of the protein.

## Figures and Tables

**Figure 1 ijms-24-06528-f001:**
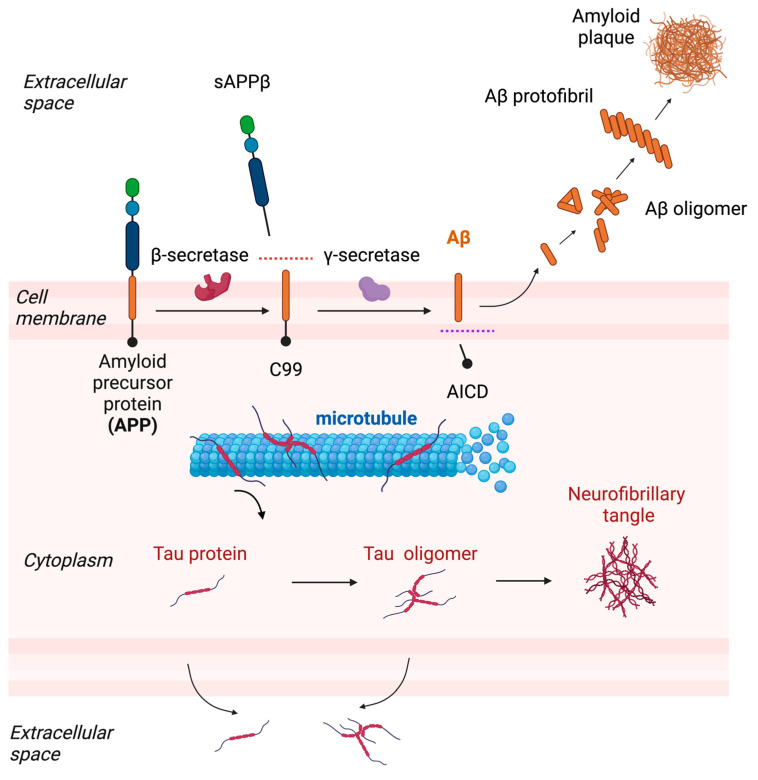
The aggregation process of Aβ and tau and their release. Aβ is generated from APP after sequential cleavages by β-secretase and γ-secretase and multimerizes into oligomer or protofibril and finally deposits as amyloid plaques. Tau is a microtubule binding protein but becomes detached from microtubules and forms oligomer and deposits as neurofibrillary tangles. Accumulating evidence has now suggested that tau is also released into the extracellular space and therefore a target of extracellular fluid clearance (adapted from “Cleavage of Amyloid Precursor Protein (APP)” and “Pathology of Alzheimer’s Disease” by Biorender.com (accessed on 28 March 2023).

**Figure 2 ijms-24-06528-f002:**
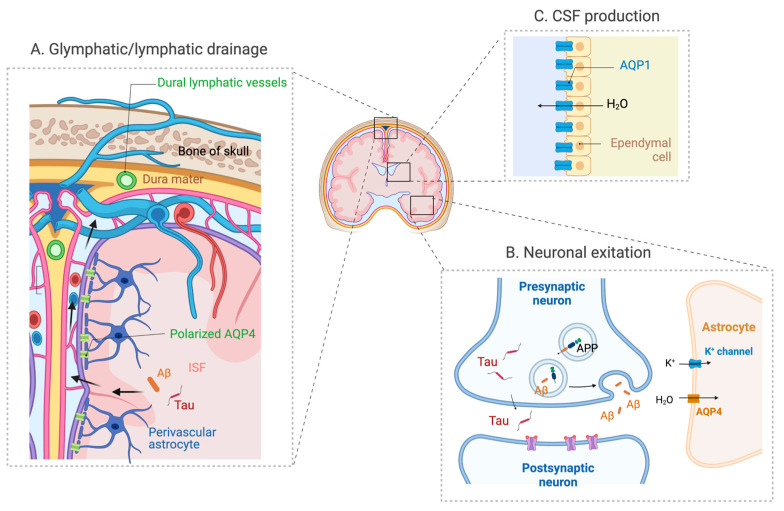
The expression and distribution of AQPs in the brain. (**A**) AQP4 expressed in perivascular astrocyte endfoot plays a significant role in the glymphatic system that drains Aβ and tau in ISF via the perivascular spaces. (**B**) AQP4 also has a role in neuronal excitation, and its deficiency may also impact the activity-dependent release of Aβ and tau. (**C**) AQP1 expressed in the choroid plexus has a role in CSF secretion (adapted from “Cranial meningeas”, “Tripartite Glutamatergic Synapse” and “Rodent Brain Subventricular Zone” by Biorender.com).

## Data Availability

Not applicable.

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
