# Peer review of "Multifaceted Roles of Aquaporins in the Pathogenesis of Alzheimer’s Disease"

_ijms, 2023, doi:10.3390/ijms24076528_

Round 1

Reviewer 1 Report

1. The introduction section should be broadened to provide a statement of the study's goals and importance. Provide a list of the materials that readers can find in this manuscript as well. Work on the abstract section.

2. Include a section outlining the importance of aquaporins for biomedical applications and their linkages to related brain illnesses.

3. Include the molecular basis of aquaporins in Alzheimer's disease and attempt to illustrate it with examples. Add a figure to explain.

4. Also look for grammatical errors.

Reviewer 2 Report

In this review article, Yamada nicely summarizes background and recent progresses regarding the  potential involvement of aquaporins on the pathogenesis of Alzheimer’s disease. Overall, the manuscript is well written with extensive references that cover both classical as well as recently published studies. The topic of this current paper has emerged rather recently and therefore many controversies as well as unanswered questions remain. In this regard, I appreciate that the author not only summarizes previous reports, but also describes limitations of current understandings in many important aspects. I believe that this review article will be of great interests and help to those who study pathophysiology of Alzheimer’s disease. I only have very minor points to suggest, as shown below.

 Line 131

“endo-foot” should be “endfoot”

Line 145

“exon4” should be “exon 4” (with space)

Line 149 and other places

“c-terminal” should be “C-terminal” (capital).

Line 158

In this section, it would be helpful to readers if one sentence is included to explain how the model was named “glymphatic system” citing the original article. In addition, while it is not necessary, I may suggest adding another sentence explaining the (existence and/or concept of) controversies regarding this model.

Lines 165-166

I feel that the statement “The best-characterized function of AQP4 in CNS to date is a modulation of glymphatic system” might be too strong. I mostly agree with the author on this point but others in the field may not, so I may suggest the author to tone down a little bit on this sentence.

Lines 273-274

The following sentence is not clear and may better be revised.

“Consistent with this, and studies from model mice have also shown that sleep deprivation elevates ISF Ab and tau levels [65,66].”

Lines 336-339

The following sentence is not clear and may better be revised.

“Second, the majority of the studies so far are suggesting that glymphatic clearance is suppressed in AD patients as well as mouse models of AD, further research is required to explore the specific molecular and cellular mechanisms underlying this process and on developing strategies to increase glymphatic flows as a potential therapeutic approach.”
